# Deciphering the Polyglucosan Accumulation Present in Lafora Disease Using an Astrocytic Cellular Model

**DOI:** 10.3390/ijms24076020

**Published:** 2023-03-23

**Authors:** Mireia Moreno-Estellés, Ángela Campos-Rodríguez, Carla Rubio-Villena, Lorena Kumarasinghe, Maria Adelaida Garcia-Gimeno, Pascual Sanz

**Affiliations:** 1Instituto de Biomedicina de Valencia (IBV-CSIC), 46010 Valencia, Spain; mmoreno@ibv.csic.es (M.M.-E.); acampos@ibv.csic.es (Á.C.-R.); lkumarasinghe@ibv.csic.es (L.K.); 2Centro de Investigación Biomédica en Red de Enfermedades Raras (CIBERER)-ISCIII, 28029 Madrid, Spain; 3Institute for Integrative Systems Biology (I2SysBio), Consejo Superior de Investigaciones Científicas (CSIC)—Universitat de València (UV), Parc Científic, Cat. Agustín Escardino 9, 46980 Paterna, Spain; carla.rubio@csic.es; 4Department of Biotechnology, Escuela Técnica Superior de Ingeniería Agronómica y del Medio Natural (ETSIAMN), Universitat Politécnica de València, 46022 Valencia, Spain; 5Consejo Superior de Investigaciones Científicas, Jaime Roig 11, 46010 Valencia, Spain

**Keywords:** Lafora disease, polyglucosan accumulation, astrocytes, disease cellular model

## Abstract

Lafora disease (LD) is a neurological disorder characterized by progressive myoclonus epilepsy. The hallmark of the disease is the presence of insoluble forms of glycogen (polyglucosan bodies, or PGBs) in the brain. The accumulation of PGBs is causative of the pathophysiological features of LD. However, despite the efforts made by different groups, the question of why PGBs accumulate in the brain is still unanswered. We have recently demonstrated that, in vivo, astrocytes accumulate most of the PGBs present in the brain, and this could lead to astrocyte dysfunction. To develop a deeper understanding of the defects present in LD astrocytes that lead to LD pathophysiology, we obtained pure primary cultures of astrocytes from LD mice from the postnatal stage under conditions that accumulate PGBs, the hallmark of LD. These cells serve as novel in vitro models for studying PGBs accumulation and related LD dysfunctions. In this sense, the metabolomics of LD astrocytes indicate that they accumulate metabolic intermediates of the upper part of the glycolytic pathway, probably as a consequence of enhanced glucose uptake. In addition, we also demonstrate the feasibility of using the model in the identification of different compounds that may reduce the accumulation of polyglucosan inclusions.

## 1. Introduction

Lafora disease (LD, OMIM#254780) is a rare and fatal neurological disorder characterized by progressive myoclonus epilepsy. The hallmark of the disease is the presence of insoluble forms of glycogen in the brain and peripheral tissues [1,2]. These glycogen-like inclusions, also known as polyglucosan bodies (PGBs), were first described in 1911 by the Spanish neurologist Dr. Gonzalo Rodriguez Lafora [1]. Affected individuals show the first symptoms during adolescence, with a marked change in behavior, depression, and dysarthria. The disease becomes more severe with time, with the appearance of myoclonic episodes, seizures, and rapid progressive neurodegeneration. A recent study on the natural history of the disease indicates that patients have a median disease duration of 11 years [3,4,5]. To date, there is no effective therapy, and patients are initially treated with anti-seizure medications, to which, after some time, they develop resistance [6].

LD is an autosomal recessive disorder. Mutations in the *EPM2A* gene, encoding Laforin, a glycan phosphatase, and the *EPM2B* gene, encoding Malin, a RING-type E3-ubiquitin ligase, are related to the disease. Both proteins work together in a complex that is involved in the regulation of glycogen synthesis, among other physiological processes (see [6] for review). To study LD pathophysiology, mouse models with deletions in the *Epm2a* or *Epm2b* genes have been generated [7,8]. Both mouse models present similar pathophysiological phenotypes, mimicking the ones present in LD patients. They show an accumulation of PGBs in the brain and peripheral tissues [8]; they are more sensitive to the effects of proconvulsant drugs such as kainate or pentylenetetrazole (PTZ) [9]; they show altered proteostasis with decreased autophagy [8,10,11]; they manifest behavioral impairments [12]; they exhibit a dysfunctional glutamate uptake, which could lead to excitotoxicity [13]; and they show reactive glia-derived neuroinflammation [14,15,16].

It has been assumed in the field that the accumulation of PGBs is a cause of the pathophysiological features of LD [17]. This assumption is supported by genetic data; in Lafora disease mouse models, by producing a decrease in the expression/activity of the enzyme that is in charge of glycogen synthesis in the brain, namely glycogen synthase 1 (GYS1), there was a reduction in the pathophysiological features of the disease [18,19,20]. For this reason, different groups have aimed to design strategies that could prevent PGB accumulation by targeting GYS1, either by using small molecule inhibitors [21], antisense oligonucleotides (ASOs) [22], or other approaches aimed at decreasing *GYS1* expression [23]. Alternatively, other authors have suggested the use of antibody-fusion enzymes coupled to alpha-amylase to degrade polyglucosan inclusions [24,25].

However, despite the efforts made by several groups during the last decade, the question of why PGBs accumulate in the brain is still unanswered. We have recently demonstrated that, in vivo, astrocytes play a key role in the pathophysiology of LD: they accumulate most of the polyglucosan inclusions (PGBs) present in the brain [26], they have defective glutamate uptake [13,27], and they produce pro-inflammatory mediators, which lead to the establishment of the characteristic neuroinflammatory landscape present in LD [15,16]. To obtain a deeper understanding of the defects present in LD mouse astrocytes that lead to the aforementioned deficiencies, we sought to obtain pure primary cultures of astrocytes from LD mice from the postnatal stage. Under specific conditions, we observed that they accumulated PGBs, serving in this way as a disease model to study PGB accumulation and related astrocyte dysfunctions.

## 2. Results

### 2.1. Obtaining a Primary Culture of Astrocytes from Control and LD Mouse Models

We use the protocol described in Materials and Methods; in brief, it comprised of the mechanical and enzymatic dissociation of the selected brain areas (cortex and hippocampus, since they are the areas most strongly related to the pathological features of LD), the magnetic depletion of microglia with anti-CD11b beads, growth in plates coated with laminin, shaking of the cell cultures to remove unattached cells (mainly residual microglia and oligodendrocytes), and maturation of the remaining cells with dibutyryl-cAMP (DB-cAMP) for 10 days (Figure 1A,B). At this stage, cells were detached from the culture plates and analyzed using cytometry with antibodies against ACSA2 (astrocytic marker), CD11b (microglia marker), and O4 (oligodendrocytic marker). We observed that more than 97% of the cells were positive for the ACSA2 astrocytic marker, indicating the high purity of astrocytes in the cell cultures (Figure 1C). In addition, we performed immunofluorescence analysis of the cultured cells using alternative specific astrocytic marker antibodies, namely glial fibrillary acidic protein (GFAP) and S-100 calcium-binding protein B (S100B). As shown in Figure 1D, most of the cells were positive for either GFAP or S100B. We observed that some cells were stained only with one of the antibodies, which is in agreement with the recently described heterogeneity of the astrocyte population, e.g., protoplasmic astrocytes produce more of the S100B marker, whereas fibrous astrocytes produce more of the GFAP marker [28,29].

### 2.2. Astrocytes from LD Mice Accumulate More Polyglucosan Inclusions Than Controls

After maturation with DB-cAMP for 10 days, purified astrocytes were cultured for 24 h in the same media containing 7.5 mM glucose but in absence of DB-cAMP to avoid any undesired effects derived from the compound. Cells were analyzed for the presence of polyglucosan inclusions using an anti-glycogen antibody. As shown in Figure 2A, astrocytes from *Epm2a−/−* and *Epm2b−/−* mice accumulated far more polyglucosan inclusions than control astrocytes in both conditions. Alternatively, we grew the cells in DMEM + 10% FBS medium containing either no extra glucose added or 25 mM glucose for 24 h, and no differences were observed between the two glucose conditions (Figure 2B). Since similar accumulation was detected in both *Epm2a−/−* and *Epm2b−/−* astrocytes, we continued the work only with *Epm2b−/−* astrocytes.

It has been noted that, in the brains of LD mice, the polyglucosan inclusions that accumulate in the animals at younger ages are sensitive to alpha-amylase treatment, whereas those inclusions present in older animals are resistant to the action of this enzyme [26,30,31]. As such, we treated the *Epm2b−/−* astrocyte cultures with alpha-amylase (diastase) and observed a significant decrease in the number of polyglucosan inclusions (Figure 2C,D). On the one hand, these results confirmed that the observed inclusions were of a carbohydrate nature, and, on the other hand, they suggested that the polyglucosan inclusions that accumulated in astrocytes obtained from the postnatal stage were sensitive to the action of alpha-amylase, which is in agreement with the sensitivity of the polyglucosan inclusions present in the young animals, as mentioned above.

### 2.3. Why Do LD Astrocytes Accumulate More Polyglucosan Inclusions?

To understand why LD astrocytes accumulated more polyglucosan inclusions, we analyzed the levels and activity of the enzymes involved in their synthesis and degradation. Although glycogen homeostasis is a complex process involving a full set of different enzymes (glycogen synthase, glycogen branching enzyme, glycogen phosphorylase, and glycogen debranching enzyme, among others) [32], we focused our attention on glycogen synthase and glycogen phosphorylase (Figure 3A). It is known that astrocytes express the muscular isoform of glycogen synthase (GYS1). This enzyme is inactivated by phosphorylation in different residues and is activated by dephosphorylation [32]. It has been previously demonstrated that there are no changes in the activity of GYS1 in either brain or muscular tissue from LD mice [33]. Consistently with these results, we observed no changes in the amount of GYS1 or its phosphorylation status in extracts from astrocytes from control or *Epm2b−/−* mice, grown for 24 h in the absence of extra glucose or in 25 mM glucose (Figure 3B). Astrocytes mainly express two isoforms of glycogen phosphorylase, the muscular (PYGM) and the brain (PYGB) isoforms, which differ in their means of becoming activated either by phosphorylation or by allosteric regulation with AMP [34,35]. We analyzed the levels of both PYGM and PYGB in *Epm2b−/−* astrocyte extracts and found no differences in comparison to the control (Figure 3C). We detected no change in the overall glycogen phosphorylase activity between *Epm2b−/−* and control astrocytes (Figure 3D), and this activity was similarly reduced in both cases in the presence of 1,4-dideoxy-1,4-imino-d-arabinitol (DAB), an inhibitor of glycogen phosphorylase [36]. These results indicated that, in *Epm2b−/−* astrocytes, there were no major differences in the studied enzymes related to glycogen synthesis and degradation, in comparison to the control cells.

Due to the absence of these changes, we decided to use a metabolomic approach to identify possible differences in intermediate metabolites, which could explain the accumulation of polyglucosan inclusions in the LD astrocytes. After maturation, control and *Epm2b−/−* astrocytes grown over 24 h in a medium with no extra glucose added were pelleted and analyzed using a global metabolomics platform at the Metabolon Company (see Material and Methods). Then, 618 independent compounds of known identity were analyzed in the samples. The levels of these compounds were normalized by Bradford protein concentration and related to those present in the control samples. Statistical significance of the relative levels of the compounds in four independent samples from each genotype was assessed using ANOVA contrasts and Welch’s two-sample t-tests (Appendix A). A summary of the biochemicals with a fold change (FC) higher than 1.30 or lower than 0.70, and with a *p*-value lower than 0.1 are indicated in Table 1. Minor changes in the levels of biochemicals were observed between *Epm2b−/−* astrocytes and the control samples. Regarding the carbohydrate metabolism, we detected an increase in the levels of Fructose-1,6 diphosphate (Fru1,6 diP) (FC 5.19; *p* = 0.023), glucose-6 phosphate (Glu-6P) (FC 3.77; *p* = 0.023), and gluconate (FC 2.27; *p* = 0.063), and a decrease in the levels of arabitol/xylitol (FC 0.58; *p* = 0.053). The rest of the significant changes were related to amino acid metabolism (1-ribosyl-5-imidazole acetate, FC 2.08; *p* = 0.027; N-methyl-GABA, FC 0.67; *p* = 0.028; 4-acetamido butanoate, FC 0.66; *p* = 0.012) and lipid metabolism (1-oleoylglycerol 18:1, FC 0.50; *p* = 0.037) (Table 1).

The higher levels of Glu-6P, Fru1,6 diP, and gluconate are consistent with our recent observation that astrocytes from *Epm2b−/−* mice have increased glucose uptake [37]. This enhanced glucose uptake may be responsible for higher levels of Glu-6P, which could improve glycogen biosynthesis (Figure 4) [32].

### 2.4. Can the Accumulation of Polyglucosan Inclusions in Epm2b−/− Astrocytes Be Prevented?

Since it is thought that the accumulation of polyglucosan inclusions is the trigger for LD pathophysiology [17], different strategies have recently been implemented to decrease the synthesis of PGBs by decreasing the expression/activity of GYS1 (see Introduction). Alternatively, we considered improving the degradation of PGBs by increasing the activity of endogenous glycogen phosphorylase (PYG) (Figure 3A). The activity of PYG is regulated by two main mechanisms, phosphorylation and allosteric activation by AMP. In the case of phosphorylation, PYG is phosphorylated by an activation cascade initiated by an increase in the levels of cAMP; this activates protein kinase A (PKA), leading to the phosphorylation of glycogen phosphorylase kinase (PHK), which is responsible for the phosphorylation of PYG (Figure 5A) [34,35]. Using the *Epm2b−/−* astrocyte cultures as a disease model, we treated them with different compounds that activate adenylate cyclase (e.g., forskolin; propranolol), but we observed no differences in the levels of polyglucosan inclusion after the treatment, suggesting that the activation of PYG by phosphorylation did not affect the degradation of polyglucosans. On the contrary, treatment for 24 h of the *Epm2b−/−* astrocytes with AICAR, an analog of AMP [38], reduced the levels of polyglucosans by 70% (Figure 5B,C). The action of AICAR was mediated through PYG activity, since no reduction in the levels of polyglucosans was detected in the presence of AICAR plus DAB (AICAR + DAB) (Figure 5B,C).

We also tested the action of metformin, a drug that, by inhibiting mitochondrial complex I, leads to an increase in the levels of endogenous AMP [39,40]. Metformin also improved the degradation of polyglucosans, although it was less effective (53% reduction) than AICAR (Figure 5D). In this case, the presence of DAB only partially reduced the capacity of metformin to reduce polyglucosan accumulation, suggesting that this drug may activate alternative routes of degradation (Figure 5B,D).

Since both AICAR and metformin are known to be activators of AMP-activated protein kinase (AMPK) [38,39], we wanted to check whether the positive action of these two drugs on the degradation of polyglucosans could be mediated by the activation of AMPK, in addition to the activation of PYG. With this aim, we treated *Epm2b−/−* astrocytes with salicylate, a direct activator of AMPK [41], and, although we observed an activation of AMPK, both as an increase in the phosphorylation status of the catalytic AMPKalpha subunit (P-AMPKalpha) and as an increase in the phosphorylation of one of its canonical substrates (acetyl-CoA carboxylase; P-ACC) (Figure 6C), no degradation of the polyglucosan inclusions was observed (Figure 6A,B). These results suggested that the activation of AMPK, by itself, played a minor role in the degradation of the polyglucosans present in *Epm2b−/−* astrocytes.

## 3. Discussion

Lafora disease (LD, OMIM254780) is a fatal form of progressive myoclonic epilepsy. The hallmark of LD is the accumulation of aberrant forms of glycogen (polyglucosans, PGBs) in the brain and peripheral tissues. Our recent work, using mouse models of LD (*Epm2a−/−* and *Epm2b−/−* mice), has demonstrated that astrocytes play an important role in the pathophysiology of LD: they accumulate polyglucosans (PGBs) [26] and they have impaired proteostasis, with decreased proteasome and autophagy activities, impaired mitochondrial function, and increased oxidative stress (see [6] for review). Moreover, they have altered glutamatergic transmission [27] and neuroinflammation, which becomes more severe as the animals grow older [15]. All these results pointed to astrocytes, the major reservoir of glycogen in the brain, playing a major role in LD. On the other hand, it has been assumed that the accumulation of PGBs is causative of the pathophysiological features of LD [17].

However, despite the efforts made by different groups during the last decade, the question of why PGBs accumulate in the brain is still without an answer. To address this question, we developed a primary culture of LD astrocytes that accumulate more PGBs than control cells when grown under specific conditions. Using this disease cellular model, we analyzed the enzymes involved in glycogen synthesis and degradation, as well as its metabolomic profile. Our results suggest that LD astrocytes accumulate more PGBs because they have higher levels of Glu-6P, possibly because of enhanced glucose uptake, as we have recently reported [37], and this could lead to glycogen accumulation (Figure 4).

This increase in the supply of Glu-6P would have a primary effect on glycogen synthesis. However, additional roles of the laforin/malin complex in the formation of polyglucosan inclusions have also been described. As laforin contains a carbohydrate-binding module (CBM20) at its N-terminus, this allows the protein to bind to polysaccharides, especially those with fewer branches, attracting malin to these sites and allowing for the ubiquitination of proteins related to glycogen metabolism such as the regulatory subunit R5/PTG of protein phosphatase 1, glycogen synthase, and others [4,6]. Therefore, the laforin/malin complex may have a dual role: one regulating glucose uptake, and another regulating glycogen synthesis (Figure 7).

The use of the LD disease cellular model described in this work has also allowed for the identification of different compounds that stimulate glycogen degradation by enhancing glycogen phosphorylase activity (e.g., AICAR, metformin). Of note is the case of metformin, a drug regularly used for the treatment of type 2 diabetes [39,40]. We reported some years ago the beneficial effects of metformin on *Epm2b−/−* mice [42], which allowed us to obtain the orphan designation of this drug for the treatment of LD by the European Medicines Agency and the Food Drug Administration from the USA. Recently, the administration of metformin from conception to adulthood was shown to improve neuronal hyperexcitability, motor and memory alterations, neurodegeneration, and astrogliosis, and to decrease the formation of polyglucosan inclusions in LD mouse models [43]. Therefore, the beneficial effect of metformin obtained in the astrocyte cellular model of LD described in this work supports the beneficial effect of this compound at the organismal level. This validates the use of the LD astrocyte cultures to identify novel compounds that could prevent the formation of PGBs or enhance their degradation.

In summary, we established a novel in vitro disease model for Lafora disease based on the accumulation of polyglucosan inclusions in astrocytes from postnatal LD mice. This model has been useful in determining that accumulation of metabolic intermediates of the upper part of the glycolytic pathway could be responsible for glycogen accumulation, probably due to enhanced glucose uptake. In addition, we have demonstrated the feasibility of the model to be used in the identification of different compounds that reduce the accumulation of polyglucosan inclusions. This model could be used as platform for pharmacological screenings of novel compounds with these properties. Therefore, the model will reduce the time required for assessing the putative beneficial effect of new drugs before entering into pre-clinical trials in LD mice.

## 4. Materials and Methods

### 4.1. Primary Mouse Astrocyte Isolation and Culture

This study was carried out according to the recommendations in the Guide for the Care and Use of Laboratory Animals of the Consejo Superior de Investigaciones Cientificas (CSIC, Madrid, Spain) and approved by the Consellería de Agricultura, Medio Ambiente, Cambio Climático y Desarrollo Rural from the Generalitat Valenciana. Mouse procedures were approved by the animal ethics committee of the Instituto de Biomedicina de Valencia-CSIC [Permit Number: IBV-51, 2019/VSC/PEA/0271]. All efforts were made to minimize animal suffering. Mouse primary astrocytes from control *Epm2a−/−* and *Epm2b−/−* mice were obtained from P0 to P1 mice. Meninges of the mouse brains were removed, cortices, including the hippocampus, were dissected, and tissues were homogenized using the Neural Tissue Dissociation kit and the GentleMACS dissociator from Miltenyi Biotec (Madrid, Spain). Once microglia were obtained, microglia depletion was performed using CD11b Microbeads (Miltenyi Biotec, Madrid, Spain). Cells were grown in Dulbecco’s modified Eagle medium (DMEM) (Lonza. Barcelona. Spain) containing 20% inactivated fetal bovine serum (FBS) (Fisher Scientific, Madrid, Spain), supplemented with 1% L-glutamine, 7.5 mM glucose, 100 units/mL penicillin, and 100 μg/mL streptomycin, in a humidified atmosphere at 37 °C with 5% of CO_2_. After 48 h, FBS was reduced to 10%. For the following 10 days, 0.25 mM dibutyryl-cAMP (DB-cAMP) (D0627, Sigma-Aldrich, Madrid, Spain) was added to the cultures to favor astrocytes’ maturation. At the end of the maturation process, primary astrocytes were grown for a further 24 h in the same media but in the absence of DB-cAMP to avoid any undesired effect deriving from the compound [44,45,46]. Alternatively, astrocytes were grown in DMEM + 10% FBS medium containing either no extra glucose or 25 mM added glucose for 24 h.

### 4.2. Cytometry

The purity of the astrocyte cultures was confirmed by flow cytometry. First, cells were blocked with FcR blocking reagent (#130-092-575, Miltenyi Biotec, Madrid, Spain,) in 1× phosphate buffer saline (PBS) and 0.5% bovine serum albumin (BSA) and then stained with combinations of fluorophore-conjugated antibodies purchased from Miltenyi Biotec: for astrocytes, ACSA-2-PE (#130-102-365); for microglia, CD11b-Vio Bright FITC (#130-109-368); and for oligodendrocytes, O4-APC (#130-095-895). Data were acquired on a FACSVERSE cytometer and analyzed using *FlowJo* v10 software (Ashland, OR, USA). Non-viable cells were excluded based on forward and side scatter profiles and 7-Aminodactinomycin (7AAD) staining (#130-111-568).

### 4.3. Immunofluorescence (IF) Analyses

Cells were fixed with 4% PFA in PBS for 15 min. Plates were immersed in blocking buffer (1% BSA, 10% FBS, 0.2% Triton X100, in PBS) and incubated overnight at 4 °C with primary antibodies diluted in a blocking buffer: anti-glycogen (a generous gift from Dr. Otto Baba; Tokyo Medical and Dental University, Tokyo, Japan), anti-GFAP (ab7260, Abcam, Madrid, Spain), and anti-S100B (ab52642, Abcam, Madrid, Spain). After 3 washes of 10 min in PBS, sections were incubated for 1 hour at room temperature with the appropriate secondary antibody diluted at 1/500 in blocking buffer, washed twice with PBS, and mounted in Fluoroshield with DAPI (F6057, Sigma, Madrid, Spain). When indicated, samples were first treated with 1 mg/mL diastase (alpha-amylase; A3176, Sigma, Madrid, Spain) in PBS for 10 min at 37 °C; then, they were washed 3 times (10 min each) with PBS and processed for immunofluorescence analyses as above. Images were acquired with a Confocal Spectral Leica TCS SP8 microscope (Leica, Wetzlar, Germany). Images were treated with the ImageJ software (Wayne Rasband, National Institutes of Health, Bethesda, MD, USA). To quantify the number of cells containing glycogen inclusions, 3 random fields at 40× were counted per condition, containing at least 100 cells in each case.

### 4.4. Western Blot Analyses

The soluble fractions of cell lysates (30 µg protein) were analyzed by SDS-PAGE and proteins were transferred to PVDF membranes (Millipore, Madrid, Spain). Membranes were blocked with 5% (*w*/*v*) non-fat milk in Tris-buffered saline Tween20 buffer (TBS-T: 50 mM Tris-HCl pH 7.4, 150 mM NaCl, 0.1% (*v*/*v*) Tween20) for 1 h at room temperature and incubated in 2% bovine serum albumin (BSA) in TBS-T overnight at 4 °C with the corresponding primary antibodies: anti-pS641-glycogen synthase (#3891, Cell Signaling, Barcelona, Spain), anti-glycogen synthase (ab40810, Abcam, Madrid, Spain), anti-PYGM (ab81901, Abcam, Madrid, Spain), anti-PYGB (ab154969, Abcam, Madrid, Spain), anti-pT172-AMPKalpha (#2535, Cell Signaling, Barcelona, Spain), anti-AMPKbeta (#4150, Cell Signaling, Barcelona, Spain), and anti-phosphoACC (#3661, Cell Signaling, Barcelona, Spain). Anti-GAPDH (sc-32233, Santa Cruz Biotechnologies, Madrid, Spain) and anti-Tubulin (T6199, Sigma-Aldrich, Madrid, Spain) were used as loading controls. After washing, membranes were incubated with the corresponding HRP-conjugated secondary antibodies (Fisher Scientific, Madrid, Spain), at 1/5000 in 2% (*w*/*v*) non-fat milk in TBS-T for 1 h at room temperature. Signals were visualized using Amersham ECL Prime Western Blotting Detection Reagent (Cytiva, Barcelona, Spain), Lumi-Light Western Blotting Substrate (Roche Applied Science, Barcelona, Spain) or ECL Prime Western Blotting Detection Reagent (GE Healthcare, Barcelona, Spain). Chemoluminiscence analysis was performed using the FujiLAS400 image reader (GE Healthcare, Barcelona, Spain).

### 4.5. Glycogen Phosphorylase Enzymatic Assay

A glycogen phosphorylase (PYG) activity assay was performed on mouse primary astrocytes (control vs. *Epm2b*−/−). As a control condition, cells were treated for 24 h with 300 μM 1.4-dideoxy 1.4 iminoarabinitol (DAB; D1542, Sigma-Aldrich, Madrid, Spain) before the assay to inhibit the activity of PYG. Cells were washed twice with cold PBS and resuspended in 350 μL of TES buffer (20 mM Tris pH 7.4, 1 mM EDTA, 225 mM sucrose, 2.5 mM DTT, 0.1 mM PMSF, and complete protease inhibitor cocktail (Roche Diagnostics, Barcelona, Spain)). Cells were lysed in cold TES buffer with a 25-gauge needle and centrifuged at 12,000× *g* for 10 min at 4 °C. Total protein (100 μg) was used to measure PYG activity in an assay buffer (50 mM K_2_H_2_PO_4_. pH 7.5, 10 mM MgCl_2_, 5 mM EDTA pH 8, 0.5 mM NADP, 1.5 units/mL glucose-6-phosphate dehydrogenase, 1 unit/mL phosphoglucomutase, and 0.1 mg/mL glycogen (all from Sigma-Aldrich, Madrid, Spain)). An assay buffer containing TES without NADP, glycogen, phosphoglucomutase, and glucose-6-phosphate dehydrogenase was added to 100 μg of total protein as a blank control. The metabolic activity assay was carried out by incubating the mixture at 37 °C for 20 min. The absorbance of the samples was detected at 340 nm in a spectrophotometer (PERKIN ELMER WALLAC VICTOR 2 Spectrophotometer). The amount of NADPH formed was determined using a standard curve of known NADPH concentrations (Sigma-Aldrich, Madrid, Spain). The activity was defined as nmols of NADPH per milligram of protein and per min.

### 4.6. Metabolomics Analyses at Metabolon Co.

#### 4.6.1. Sample collection

After maturation, astrocytes were grown in DMEM + 10% FBS medium with no extra added glucose for 24 h. Then they were gently trypsinized, washed by centrifugation, counted, and collected by flash-freezing. Packed cell pellets of approximately of 100 µL were stored at −80 °C until shipment.

#### 4.6.2. Sample Preparation and Analysis

Samples were sent to Metabolon, Inc. (Durham, NC, USA) for metabolomic profiling studies. Briefly, astrocytes pellets were prepared using the automated MicroLab STAR^®^ system from Hamilton Company. To remove the protein, small molecules bound to protein or trapped in the precipitated protein matrix were dissociated; to recover chemically diverse metabolites, proteins were precipitated with methanol under vigorous shaking for 2 min (GlenMills GenoGrinder 2000; GlenMills, Clifton, NJ, USA), followed by centrifugation. The resulting extract was divided into five fractions: two for analysis by two separate reverse phase (RP)/UPLC-MS/MS methods with positive ion mode electrospray ionization (ESI), one for analysis by RP/UPLC-MS/MS with negative ion mode ESI, one for analysis by HILIC/UPLC-MS/MS with negative ion mode ESI, and one sample was reserved for backup. Samples were placed briefly on a TurboVap^®^ (Zymark; Bimedis, FL, USA) to remove the organic solvent. Peaks were quantified using the area under the curve. Only those biochemicals with a false discovery rate lower than 0.10 are shown in Table 1. A full description of the values of the 618 analyzed metabolites is presented in Appendix A.

### 4.7. Statistical Analysis

Results are shown as means +/− standard error of the mean (SEM) of at least three independent experiments. Differences between samples were analyzed using unpaired two-tailed Student’s *t*-tests using Graph Pad Prism version 5.0 statistical software (La Jolla, CA, USA). *p*-values were considered as * *p* < 0.05, ** *p* < 0.01, and *** *p* < 0.001.

## Figures and Tables

**Figure 1 ijms-24-06020-f001:**
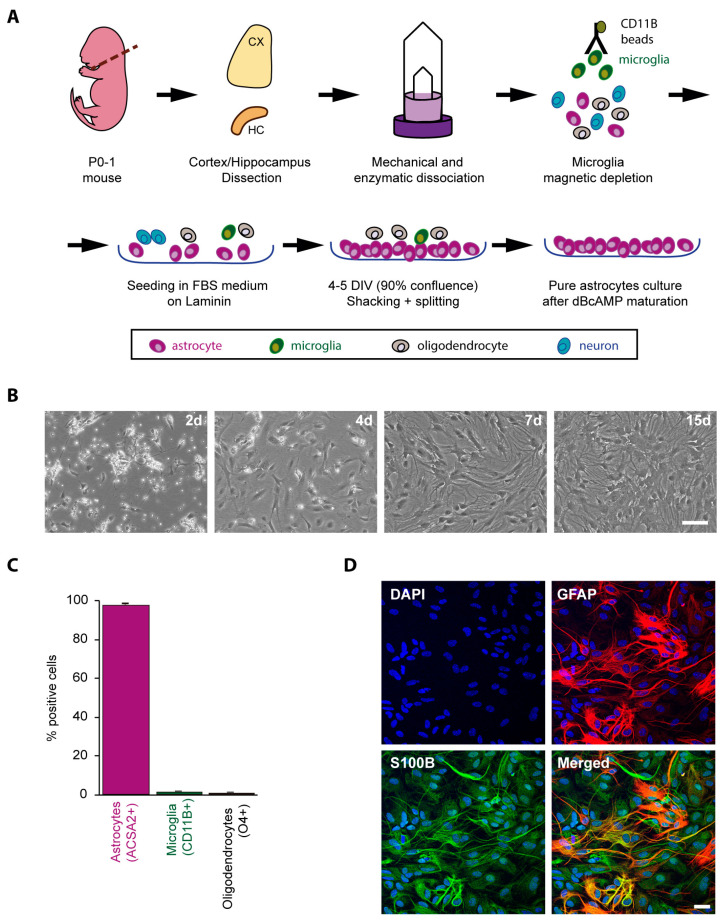
Obtaining a purified postnatal primary astrocyte culture. (**A**) Scheme of the protocol for obtaining purified astrocyte cultures (CX, cortex; HC, hippocampus; DIV, days of culture in vitro). (**B**) Pictures of the astrocytes at different days of culture. Bar indicates 100 µm. (**C**) Cytometry analyses of purified astrocytes after 10 days of maturation with DB-cAMP. The percentage of cells that were positive for anti-ACSA2 (astrocyte marker), anti-CD11b (microglia marker), and anti-O4 (oligodendrocyte marker) are indicated. Results are the mean of three independent experiments. (**D**) Immunofluorescence analyses of purified astrocytes. A representative immunofluorescence image of primary astrocyte cultures stained with anti-GFAP (red) and S100B (green) is presented. Nuclei were stained using DAPI (blue). Bar indicates 25 µm.

**Figure 2 ijms-24-06020-f002:**
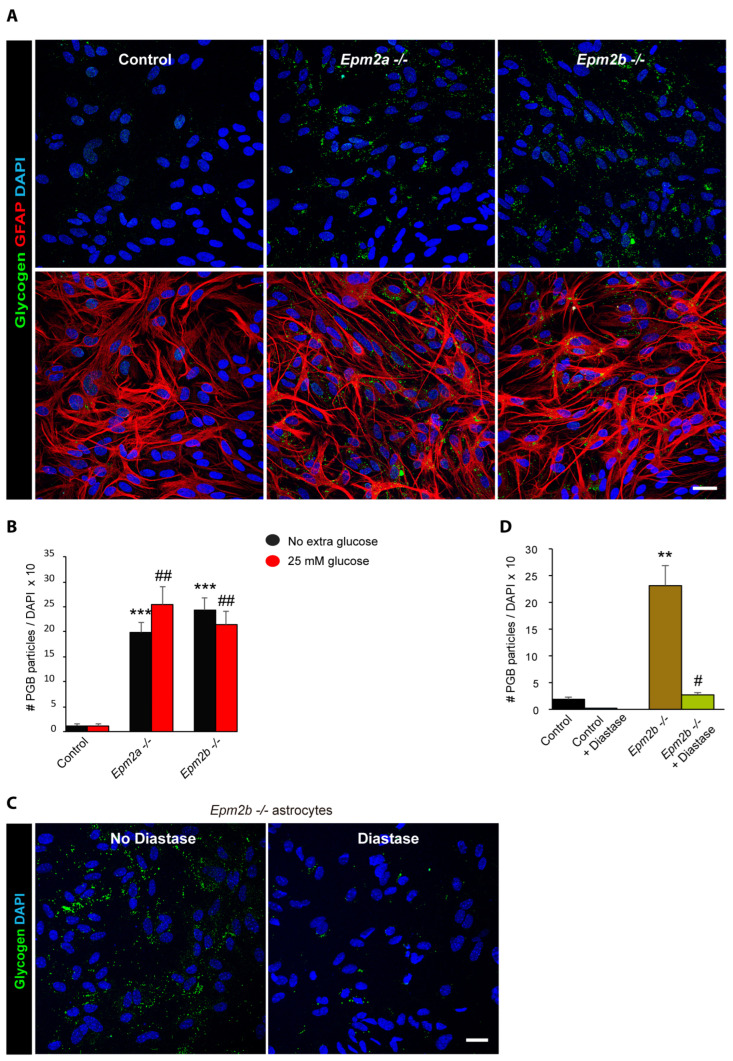
Primary astrocytes from LD mouse models accumulate polyglucosan inclusions (PGBs). (**A**) Astrocyte primary cultures from control *Epm2a−/−* and *Epm2b−/−* mice grown in DMEM + 10% FBS medium with 7.5 mM glucose for 24 h were obtained and analyzed with immunofluorescence using anti-glycogen (green) and anti-GFAP (red) antibodies. Nuclei were stained with DAPI (blue). Bar indicates 25 µm. (**B**) Quantification of the number of PGBs per cell in astrocytes grown after maturation during 24 h in media containing either no extra glucose or 25 mM glucose. Values are the mean +/− SEM of at least six independent experiments (*** *p* < 0.001, LD astrocytes with respect to the control in DMEM + 10% FBS medium with no extra glucose added; ## *p* < 0.01, LD astrocytes with respect to the control in 25 mM glucose). (**C**) PGBs accumulated in LD astrocytes are sensitive to diastase treatment. Representative images of *Epm2b−/−* astrocyte cultures treated with 1 mg/mL diastase for 10 min at 37 °C and processed by immunofluorescence with an anti-glycogen (green) antibody and DAPI (blue) are shown. Bar indicates 25 µm. (**D**) Quantification of the number of PGBs per cell in astrocytes after diastase treatment in control and *Epm2b−/−* astrocytes. Values are the mean +/− SEM of at least three independent experiments (** *p* < 0.01, *Epm2b−/−* astrocytes with respect to the control; # *p* < 0.05, *Epm2b−/−* astrocytes with respect to untreated cells).

**Figure 3 ijms-24-06020-f003:**
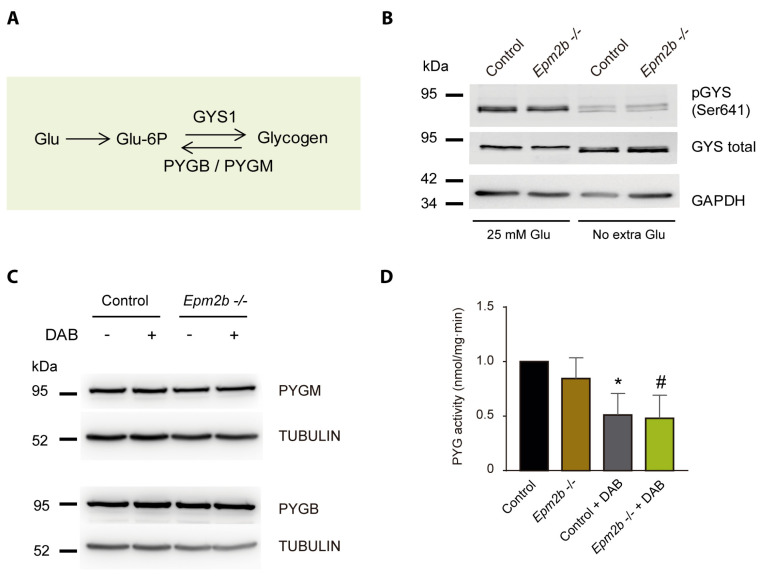
Glycogen synthase and glycogen phosphorylase are not affected in *Epm2b−/−* astrocytes. (**A**) Diagram of the synthesis of glycogen mediated by glycogen synthase (GYS1) and the two isoforms of glycogen phosphorylase (PYGB and PYGM) present in astrocytes. Glu, glucose; Glu-6P, glucose 6-phosphate. (**B**) Western blot analysis of the levels of glycogen synthase in astrocytes. Crude extracts from control and *Epm2b−/−* astrocytes grown in a medium containing 25 mM or one with no extra glucose added were analyzed using anti-GYS1, anti-phospho-Ser641-GYS1, and anti-GADPH antibodies (loading control). A representative image of three independent experiments is shown. Molecular size markers are indicated on the left. (**C**) Western blot analysis of the levels of the two major glycogen phosphorylase isoforms in astrocytes. Crude extracts from control and *Epm2b−/−* astrocytes grown in a medium with no extra glucose added were analyzed using anti-PYGM, anti-PYGB, and anti-tubulin antibodies (loading control). When indicated, cells were treated for 24 h with 300 µM of 1.4-dideoxy 1.4 iminoarabinitol (DAB), an inhibitor of glycogen phosphorylase. A representative image of three independent experiments is shown. Molecular size markers are indicated on the left. (**D**) Quantification of the total glycogen phosphorylase activity in astrocyte cultures. Glycogen phosphorylase activity was measured as described in Materials and Methods in control and *Epm2b−/−* astrocytes grown in a medium with no extra glucose added. When indicated, aliquots of these cells were treated with 300 µM of 1.4-dideoxy 1.4 iminoarabinitol (DAB) for 24 h. Values are the mean +/− SEM of at least 3 independent experiments (* *p* < 0.05, control astrocytes with respect to untreated control cells; # *p* < 0.05, *Epm2b−/−* astrocytes with respect to untreated cells).

**Figure 4 ijms-24-06020-f004:**
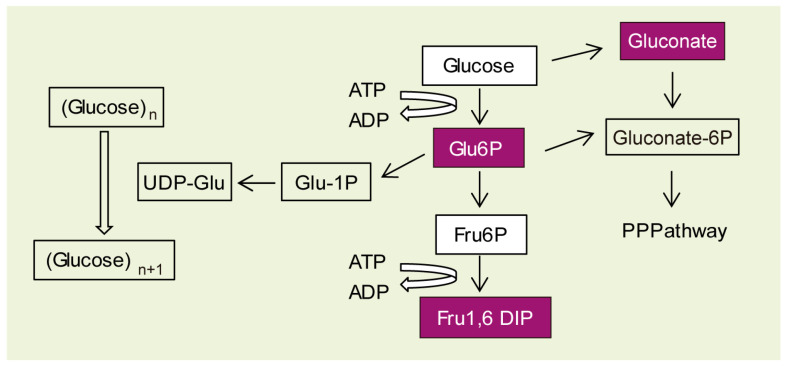
Scheme of the upper part of the metabolism of glucose. Metabolites present in higher levels in *Epm2b−/−* astrocytes in comparison to controls are shaded in purple (see Table 1).

**Figure 5 ijms-24-06020-f005:**
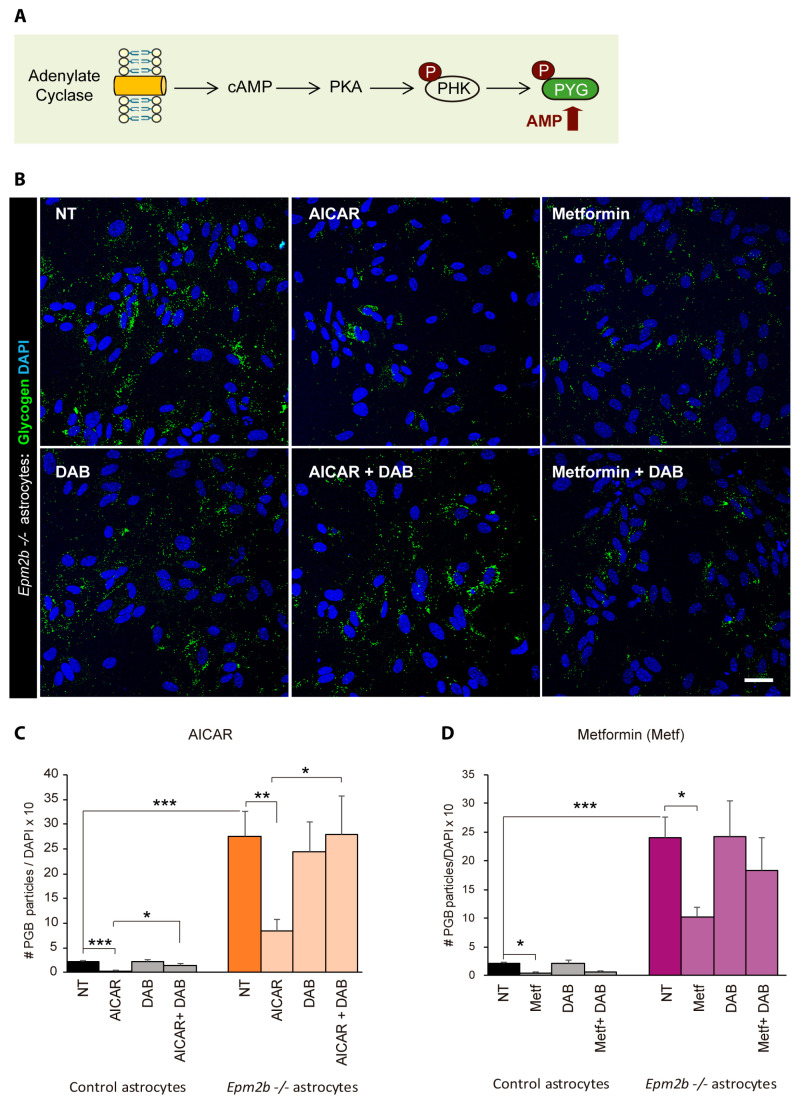
Treatment of *Epm2b−/−* astrocytes with AICAR and metformin induces the degradation of PGBs. (**A**) Scheme of the two possible forms of activation of glycogen phosphorylase: phosphorylation by the cAMP-PKA-PHK pathway and by allosteric activation by AMP. PKA, protein kinase A; PHK, phosphorylase kinase; PYG, glycogen phosphorylase. (**B**) *Epm2b−/−* astrocytes grown in a medium with 7.5 mM glucose for 24 h were either treated with 2 mM AICAR or 2 mM metformin for 24 h or were untreated. Then, the accumulation of PGBs was assessed by immunofluorescence, where glycogen is stained in green and nuclei in blue. Alternative cells were treated with 300 µM DAB or a combination of 2 mM AICAR plus 300 µM DAB (AICAR +DAB) or 2 mM metformin plus 300 µM DAB (Metf + DAB) for 24 h. Bar indicates 25 µm. (**C**,**D**) Quantification of the number of PGBs per cell in AICAR- and Metformin-treated cells, respectively. Values are the mean +/− SEM of at least six independent experiments (* *p* < 0.05; ** *p* < 0.01; *** *p* < 0.001).

**Figure 6 ijms-24-06020-f006:**
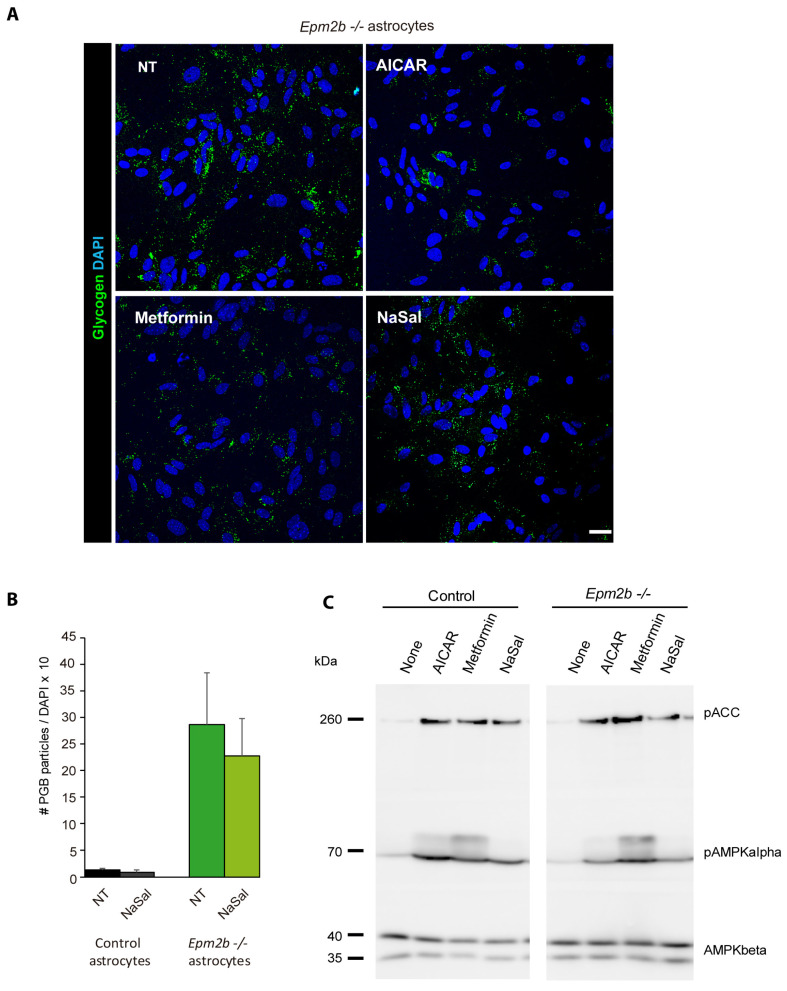
Treatment of *Epm2b−/−* astrocytes with salicylate (a direct activator of AMP-activated protein kinase) does not induce the degradation of PGBs. (**A**) Comparative panel of *Epm2b−/−* astrocytes treated with 2 mM AICAR, 2 mM metformin, or 15 mM salicylate for 24 h. The accumulation of PGBs was assessed by immunofluorescence (glycogen in green and nuclei in blue). Bar indicates 25 µm. (**B**) Quantification of the number of PGBs per cell in astrocytes treated with 15 mM salicylate. Values are the mean +/− SEM of at least three independent experiments. (**C**) Western blot analysis of control and *Epm2b−/−* astrocytes treated or not treated with the indicated compounds for 24 h. Crude extracts were analyzed using anti-phospho-ACC, anti-phospho-AMPKalpha, and anti-AMPKbeta antibodies (loading control). Molecular size markers are indicated on the left.

**Figure 7 ijms-24-06020-f007:**
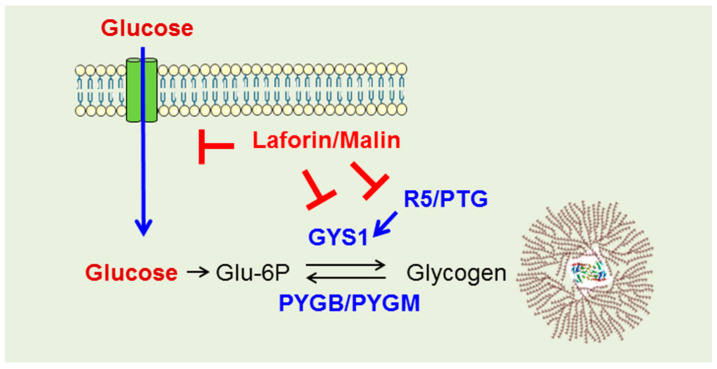
Diagram of the dual action of the laforin/malin complex on the regulation of glucose entry and glycogen synthesis. R5/PTG, regulatory subunit of protein phosphatase 1, GYS1, glycogen synthase; PYGB/PYGM, brain and muscular isoforms of glycogen phosphorylase.

**Table 1 ijms-24-06020-t001:** Differential changes in the levels of intermediate metabolites in control and *Epm2b−/−* astrocytes grown after maturation for 24 h in DMEM + 10% FBS medium with no extra glucose added. Mean relative levels of metabolites with a fold change (FC) higher than 1.30 or lower than 0.70 and with a false discovery rate and *p*-values lower than 0.1 are indicated (*n* = 4) (see Appendix A).

		*Epm2b−/−* vs. Control	
Biochemical Pathway	Metabolite	FC	*p*-Value
**UP**			
Carbohydrate metabolism	Fru-1,6 diP	5.19	0.023
	Glu-6P	3.77	0.023
	Gluconate	2.27	0.063
Amino acid metabolism (His)	1-ribosyl-5-imidazole acetate	2.08	0.027
Nucleotide metabolism (Purines)	2-deoxyguanosine	1.66	0.080
**DOWN**			
Carbohydrate metabolism	arabitol/xylitol	0.58	0.053
Amino acid metabolism (Glu)	N-methyl-GABA	0.67	0.028
Amino acid metabolism (Polyamines)	4-Acetamido butanoate	0.66	0.012
Peptides (dipeptides)	Leucylalanine	0.48	0.085
Primary bile acid metabolism	Taurocholate	0.60	0.087
Benzoate metabolism	3-hydroxy hippurate	0.55	0.060
Lipid metabolism (Monoacylglycerol)	1-oleoylglycerol (18:1)	0.50	0.037

## Data Availability

The data presented in this work will be available upon reasoned request.

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
