# Peer review of "Deciphering the Polyglucosan Accumulation Present in Lafora Disease Using an Astrocytic Cellular Model"

_ijms, 2023, doi:10.3390/ijms24076020_

Round 1

Reviewer 1 Report

Comments to author

Manuscript number ijms-2274377

Manuscript title: Deciphering the polyglucosan accumulation present in Lafora disease by using an astrocytic cellular model is a good piece of work by the author

Overall, the paper is well written and the quality of this manuscript is good for publication but in ‘Introduction’ and ‘Materials and Method’ sections are need to be rewrite and reduce plagiarism (Plagiarism report attached).

My recommendation is therefore may accept after re-orientation.

Author Response

We want to thank this reviewer for his/her positive statement on our work. Following his/her suggestions we have minimize the auto-plagiarism that has been detected in the manuscript. We have focused our attention mainly to those paragraphs with more than 1% plagiarism detected at the Introduction and Materials and Methods.

Reviewer 2 Report

The study by Moreno-Estellés on “Deciphering the polyglucosan accumulation present in Lafora disease by using an astrocytic cellular model” is interesting. The authors presented metabolomics of LD astrocytes, indicating the accumulation of metabolic intermediates of the upper part of the glycolytic pathway, probably as a consequence of enhanced glucose uptake. In addition, they also demonstrate the feasibility of the model to be used in the identification of different compounds, which may reduce the accumulation of polyglucosan inclusions.

I would suggested authors comment on structural domains in laforin, CBM20 and dual specificity phosphatase domain as well as structural domains in malin, RING and NHL (NCL1, HT2A and LIN-41 containing) on the accumulation of PGBs.

Role of malin ubiquitination on the accumulation of PGBs.

Role of R5/PTG (PPP1R3C) and glutamate transporter on the accumulation of PGBs.

I would suggest including more summarized message and a figure summarizing the mechanism.

Author Response

We want to thank the reviewer very much for his/her positive statement on our work. We have added the following paragraph to the Discussion: “This increase in the supply of Glu-6P would be a primary effect on glycogen synthesis. However, it has also been described additional roles of the laforin/malin complex in the formation of polyglucosan inclusions. As laforin contains a carbohydrate binding module (CBM20) at its N-terminus, this allows the protein to bind to polysaccharides, especially those with less branches, attracting Malin to these sites and allowing the ubiquitination of proteins related to glycogen metabolism such as the regulatory subunit R5/PTG of protein phosphatase 1, glycogen synthase and others [4], [6]. Therefore, the laforin/malin complex may have a dual role: one regulating glucose uptake, and another regulating glycogen synthesis (Figure 7)”. We have also added an additional diagram summarizing the mechanism.

Reviewer 3 Report

The topic of the study is relevant in the field. It addresses a novelty by setting up a novel in vitro disease model for Lafora disease based on the accumulation of polyglucosan inclusions in astrocytes from postnatal LD mice, which could be used as a platform for pharmacological screenings of novel compounds for their ability to reduce the accumulation of polyglucosan inclusions. In addition, it contributes to understanding the mechanisms underlying the accumulation of polyglucosan bodies in the brain. The methodology is clear, and the results are well presented and discussed.

Author Response

We want to thank the reviewer very much for his/her positive statement on our work.